# Potential Clinically Relevant Effects of Sialylation on Human Serum AAG-Drug Interactions Assessed by Isothermal Titration Calorimetry: Insight into Pharmacoglycomics?

**DOI:** 10.3390/ijms24108472

**Published:** 2023-05-09

**Authors:** Robert Kerep, Tino Šeba, Valentina Borko, Tin Weitner, Toma Keser, Gordan Lauc, Mario Gabričević

**Affiliations:** 1Department of General and Inorganic Chemistry, Faculty of Pharmacy and Biochemistry, University of Zagreb, 10000 Zagreb, Croatia; 2Department of Biochemistry and Molecular Biology, Faculty of Pharmacy and Biochemistry, University of Zagreb, 10000 Zagreb, Croatia

**Keywords:** alpha-1 acid glycoprotein, drugs, plasma protein binding, sialic acid, binding affinity, isothermal titration calorimetry

## Abstract

Human serum alpha-1 acid glycoprotein is an acute-phase plasma protein involved in the binding and transport of many drugs, especially basic and lipophilic substances. It has been reported that the sialic acid groups that terminate the N–glycan chains of alpha-1 acid glycoprotein change in response to certain health conditions and may have a major impact on drug binding to alpha-1 acid glycoprotein. The interaction between native or desialylated alpha-1 acid glycoprotein and four representative drugs—clindamycin, diltiazem, lidocaine, and warfarin—was quantitatively evaluated using isothermal titration calorimetry. The calorimetry assay used here is a convenient and widely used approach to directly measure the amount of heat released or absorbed during the association processes of biomolecules in solution and to quantitatively estimate the thermodynamics of the interaction. The results showed that the binding of drugs with alpha-1 acid glycoprotein were enthalpy-driven exothermic interactions, and the binding affinity was in the range of 10^−5^–10^−6^ M. Desialylated alpha-1 acid glycoprotein showed significantly different binding with diltiazem, lidocaine, and warfarin compared with native alpha-1 acid glycoprotein, whereas clindamycin showed no significant difference. Therefore, a different degree of sialylation may result in different binding affinities, and the clinical significance of changes in sialylation or glycosylation of alpha-1 acid glycoprotein in general should not be neglected.

## 1. Introduction

The interactions of drugs and proteins in the circulation can influence the fate of the drug in the organism, including transport, distribution, metabolism, and excretion [1,2]. Plasma protein binding of drugs is often the first step in drug distribution and as such is increasingly being studied. According to the Free Drug Hypothesis, only the free or unbound drug is available to act at physiological sites of action. Plasma protein binding may influence pharmacokinetic properties such as free fraction, clearance and volume of distribution [3]. 

Alpha-1 acid glycoprotein (AAG), also known as orosomucoid (ORM), is an abundant human plasma protein (1–3%) whose biological role remains unclear. AAG binds endogenous and exogenous ligands and transports them to target sites [4,5]. Its contribution to plasma protein binding has historically been underestimated due to its lower plasma concentration compared to albumin. Although AAG is a minor component of plasma compared to albumin, it is one of major plasma proteins responsible for binding basic and neutral ligands like drugs, serotonin and platelet activating factor [4,5,6,7,8,9,10,11].

Human AAG contains 183 amino acids with an average molar mass of 42 kDa and is highly glycosylated containing up to five N–linked glycan chains (Appendix A) [10,12]. There are two main genetic variants (F1/S and A), whose expression is under the control of two adjacent genes ORM1 (encodes F1/S) and ORM2 (encodes A) [5,13,14], that show different binding affinity for some ligands [15]. The difference between F1/S and A variants is due to 22 amino acid substitutions that affect the affinity and stereoselectivity of drug binding to AAG [8,13,16,17]. AAG is composed of 59% peptide residues and 41% carbohydrates, of which approximately 11% are sialic acids. This high sialic acid content contributes to the negative charge on its surface [5,18] and consequently to a low isoelectric point (2.8–3.8) [10]. 

In addition to genetic variants, differences in the binding of drugs to AAG may also be related to the alternate glycosylation, including the varying content of sialic acid on AAG [19,20,21]. Since the glycan structures show microheterogeneity under physiological conditions, the fully or partially desialylated AAG has been reported to be present in plasma of patients with certain diseases such as cancer, liver cirrhosis, and inflammatory rheumatic disease [22,23]. Composition of the AAG glycome have also been reported to change significantly in disease [24], suggesting that in addition to inter-individual variation, there are also temporal changes in AAG glycosylation within an individual, which may affect bioavailability of different drugs. Until recently, the analysis of differences in glycosylation was challenging, but the progress in high throughput glycomics [25,26] is making pharmacoglycomics a realistic goal. 

AAG is one of the acute phase serum proteins, and its concentration can be altered in numerous physiological and pathological conditions like injuries, inflammation, infections, pregnancy etc. [27,28,29]. Depending on conditions, concentration may increase two- to six-fold [30]. AAG is also identified as one of the potential biomarkers for assessing the risk of all-cause mortality along with albumin, very low-density lipoprotein, and citrate [31]. A decrease in AAG plasma levels has also been confirmed in females due to a suppressive effect of estrogen on AAG expression [32,33]. Newborn infants have lower AAG levels than adults which rapidly increase during the first month of life and continue rising during the first year [4]. Such variation of AAG concentration in plasma under different conditions could provide challenges in the administration of the correct dose of drugs, particularly for one with a narrow therapeutic index and high affinity for AAG. Therefore, it is essential to understand the influence of AAG’s concentration and microheterogeneity on free drug concentration in personalized medicine. 

To the best of our knowledge, this is one of the first studies focused on the influence of AAG’s sialic acid residues on the binding of drugs. The binding affinity and thermodynamic characterization of the interactions between different AAG sialo forms with clindamycin, diltiazem, lidocaine and warfarin was determined using the isothermal titration calorimetry (ITC). 

## 2. Results and Discussion

This work uses one of the best methods for obtaining the thermodynamic parameters (ITC) on AAG–drug interactions to provide insight into sialylation-based changes in the stability complex of AAG and four different drugs (clindamycin, diltiazem, lidocaine and warfarin). These sialylation changes may influence protein–drug complex stability and consequently pharmacokinetics. The investigated drugs were selected because they have relatively high binding affinity for AAG and good solubility in water at pH 7.40, which is critical for mimicking physiological conditions. In addition, binding to plasma proteins can have a major impact on the free drug fraction due to the low volume of distribution and narrow therapeutic window [3].

### 2.1. Enzymatic Desialylation of Human AAG

AAG has five N–glycan residues that have di-, tri-, and/or tetraantennary structures, with sialic acid as the terminal group. To calculate index of sialylation (IS), as defined elsewhere [34], the abundance of each N-glycan structure on the AAG variants was quantified. The corresponding UPLC–FLD peak area of each glycan identified in the crude AAG samples was normalized to the total glycan peak area (Figure 1). The index of sialylation for native and desialylated AAG (AAG+s and AAG−s, respectively) confirmed the result of desialylation with a 93.9% reduction in the sialic acid content of AAG. The percentage of each glycan in the total integral and the index of sialylation for AAG+s and AAG−s is shown in Appendix A [35]. 

### 2.2. Analysis of Binding Affinity to Native and Desialylated AAG

The evaluation of protein–drug interactions is an important factor in drug development. ITC is the only technique that allows full thermodynamic characterization in a single experiment to estimate the driving forces that characterize protein–drug interactions. Figure 2 shows typical titration curves representing an exothermic binding reaction observed for all interactions studied. The binding constants (equilibrium dissociation constants) determined from the data presented in Figure 2 are summarized in Table 1, while the binding constants (logarithmic scale) are shown in Appendix A and Appendix A. The shape of the ITC curve depends on the C value, as shown in Equation (1):(1)$$C=N\;\times \;[\mathrm{titrand}]/K_{D}$$
where *N* is the number of drug binding site(s) on the protein, *K*_D_ is the binding affinity and [titrand] is the total concentration of protein. The C values of the titrations of native AAG ranged from C = 16.9 for lidocaine (strongest binding), C = 12.2 for diltiazem, C = 8.33 for warfarin to C = 4.90 for clindamycin (weakest binding), whereas binding to desialylated AAG ranged from C = 96.9 for warfarin, C = 31.4 for lidocaine, C = 22.4 for diltiazem and C = 3.32 for clindamycin. The local anesthetic lidocaine and the anticoagulant warfarin showed the highest binding to native AAG and desialylated AAG, respectively. In both cases, clindamycin showed the lowest affinity. Data for all interactions were fitted to a single binding site model. Considering that the AAG+s and AAG−s samples are in fact mixtures of various glycoforms (Figure 1), the obtained binding affinity results are weighted averages of binding to these variants. Therefore, any influence observed should come only from the difference in sialic acid content, since the measurements are always performed with the same combination of glycoforms, either with or without terminal sialic acids.

### 2.3. Thermodynamic Analysis of Drug Binding for the Native and Desialylated AAG

Protein–drug interactions involve specific non-covalent interactions with a particular region in the protein called the binding site. When the Gibbs free energy is negative, the process is spontaneous, i.e., binding occurs. The ITC experiments in this study showed a negative deviation of the signal from baseline at each injection, indicating that heat was released, i.e., that the binding reactions of AAG with drugs were exothermic (Figure 2A,B). A return to baseline indicates that equilibrium has been established. In the lower part of Figure 2, the titrations are shown as isotherms indicating the integrated thermal responses normalized to the amount of drug injected. The thermodynamic profiles obtained with the PEAQ–ITC Analysis Software 1.30 are summarized in Table 2 and presented below.

### 2.4. Driving Forces for Drug Binding to AAG

The Gibbs free energy consists of two contributions, the enthalpy and the entropy change according to the basic equation of thermodynamics:(2)$$\Delta_{r}G{}^{\circ}=\Delta_{r}H{}^{\circ}-T\Delta_{r}S{}^{\circ}$$
higher binding affinity is usually achieved when enthalpy and entropy favorably contribute to the binding. In this study, the values of $\Delta_{r}$*G*° are negative for all reactions, indicating spontaneous formation of a protein–drug complex for both native and desialylated AAG. The data show greater stability of the AAG−s complexes, except in the case of clindamycin, where there is no statistical difference in the binding constant compared to the native AAG complex (Appendix A).

The binding of clindamycin and diltiazem has a similar thermodynamic profile for both native and desialylated AAG. Relatively small differences in the $\Delta_{r}$*G*° and the $\Delta_{r}$*H*° (AAG+s vs. AAG−s) values indicate a more dominant effect of entropy on binding, likely resulting from non-covalent binding to the apolar surface area of the protein [36]. Such binding always affects the solvent, as hydrogen bonds with both the protein and the drug are reorganized after binding occurs. Clindamycin and diltiazem are present in partially protonated form (about 30%) at the measured pH [37,38], which may result in an unfavorable entropy change in the absence of the negatively charged sialic acids in AAG−s. However, entropic contribution due to the binding of the mentioned drugs in the hydrophobic groove of AAG is not just a result of smaller degrees of freedom upon binding, but also a result of the release of water molecules during the desolvation of the reaction participants which may overcome the effect of lack of sialic acid charge influence.

The binding of lidocaine is enthalpy driven and showed no statistically significant difference (*p* = 0.336) in the $\Delta_{r}$*H*° values for AAG+s and AAG−s. It has been reported that the β-sheet of AAG, which forms the ligand binding cavity, is structurally stabilized upon ligand binding and that desialylation induces conformational changes of AAG that stabilize two regions (residues 9–18 and 80–89) located outside the β-sheet of AAG (Appendix A) [39]. It seems that lidocaine binds more strongly to AAG−s not only due to the higher contribution of hydrophobic interactions but also due to the change in entropy. Why entropy is so favorable in this case and not in the other two cases with similar p*K*_A_ [38] values remains to be discussed. First, the favorable entropy reflects the natural tendency toward disordered behavior that usually results from perturbations in solvation interactions and the simultaneous release of bound water, which appears to be more abundant for binding to AAG−s. Second, lidocaine has a positive charge on the tertiary amine with a diethylamino group that is more flexible (Appendix A). The absence of negatively charged sialic acid residues near the positive, flexible lidocaine diethylamino moiety could increase the overall entropy due to differences in overall hydrogen bonding with the solvent compared with the native protein (AAG+s).

The binding of warfarin to AAG showed a markedly different thermodynamic profile than that of other drugs. In this case, we observed strongly negative values for enthalpy and entropy. The reaction is strongly enthalpy driven and overwhelms the unfavorable entropy change. Because of the warfarin structure (hydrophobic coumarin ring, Appendix A), we can expect strong intermolecular forces with the apolar surface of AAG and possible hydrogen bonding of the negatively charged warfarin (more than 99%) at measured pH [40]. Such binding would significantly reduce the degrees of freedom and lead to high negative entropy. After removal of the sialic acids, an increase in the stability of the protein–drug complex is observed for AAG−s. Considering a small change in enthalpy compared to native AAG, this increased stability is mainly due to lowering of the strong negative entropy contribution. Removal of sialic acids may decrease repulsion with negatively charged warfarin and consequently decrease entropy, but it seems that in this case entropy increase due to rearrangement of water molecules overcomes the lowering of charge repulsion effect in addition to the increased hydrophobicity [41].

The binding enthalpies reported in this study are based solely on titrations performed in a single buffer, without correction for enthalpies associated with protonation/deprotonation [42]. As a result, the obtained thermodynamic parameters reflect not only the binding of the drugs with AAG but also the exchange of protons with the buffer (if any). It is likely that the observed thermodynamic parameters will vary in different buffers, with enthalpy values differing relative to the protonation/deprotonation enthalpy of HEPES. Additional titrations can be performed in multiple buffers, and the number of protons exchanged could be calculated by linearization methods [42,43] but those measurements are beyond the main scope of this paper.

### 2.5. Determination of the Free Fraction of Drugs

AAG is an acute-phase protein with variable concentration in plasma and this consequently affects the free drug concentration. Hyperquad simulation and speciation software (HySS 4.0.31) [44] was used to calculate the fraction of free vs. bound drug by solving the mass balance equations using the measured binding affinities. The usual peak and trough therapeutic plasma levels, *C*_max_ and *C*_trough_, were chosen (Table 3), while the low (12 μM), medium (22 μM), high (31 μM), and acute (69 μM) plasma levels were used for the AAG concentration [45]. The plasma concentrations of the drug vary over time depending on the dosing regimen, and steady–state concentrations were used in the model.

The results of the determined free fraction of drugs in both cases (native and desialylated AAG) depending on the peak and trough therapeutic levels can be seen in Appendix A. In all cases, the free fraction decreases significantly with increasing AAG concentration. Even more revealing is Figure 3 below, which shows the percent difference in free drug (AAG+s vs. AAG−s) as a function of *C*_max_ and *C*_trough_ concentrations. The free fraction of clindamycin reaching peak plasma levels was not significantly different between AAG+s and AAG−s. For diltiazem and lidocaine, the free plasma concentration decreased by approximately 20–30% with AAG−s, whereas the free plasma concentration of warfarin decreased almost five-fold. In addition to AAG, human serum albumin (HSA) also plays a critical role in drug binding due to its high plasma concentration. Therefore, to obtain more realistic estimates, the free drug concentration was calculated taking into account the available data on the binding of the studied drugs to HSA. Since the normal concentration range of HSA is between 35–50 mg/mL [50], the average of this range is 42.5 mg/mL, which corresponds to 0.64 mM.

The results presented in Figure 4 below show that the free drug concentration is significantly decreased by the AAG concentration only in the case of lidocaine, due to its very weak albumin binding, whereas the other drugs show a small increase in the free drug concentration, indicating a strong “buffering capacity” of albumin for these drugs. However, when we calculate the difference between binding to AAG+s and AAG−s in the presence of has, the concentration of free drug decreases by about 10% to 40% for lidocaine and warfarin, depending on the conditions (Figure 5).

### 2.6. Pharmacoglycomics

Advances in genetic engineering have enabled the development of pharmacogenomics, a field that has highlighted the importance of interindividual variation in gene sequence for the safe use of many drugs. However, gene sequence is only one aspect of variation in protein structure and function. Glycosylation is the highest level of molecular complexity that enriches the structure and function of most proteins. The main feature of glycosylation is that it is not controlled by a template. Instead of being encoded by a single gene, glycans are encoded by a complex network of genes, their epigenetic regulation, and the environment [51,52]. Multiple genes are involved in regulation of protein glycosylation [53] and these regulatory networks are protein-specific [54]. Restructuring of gene variants in these complex networks causes proteins to differ both structurally and functionally [55]. In this study, we have shown that changes in the sialylation of AAG can have significant effects on its ability to bind common drugs. Since previous studies demonstrated that AAG glycosylation can significantly change in different diseases, this indicates that carefully titrated drugs like warfarin might come out of their therapeutic range because of changes in AAG glycosylation. 

As far as we can tell, this is one of the first studies to consider not only changes in AAG content but also changes in glycosylation pattern and albumin binding when calculating free drug concentration. Wu et al. [56] recently measured the effects of AAG glycosylation microheterogeneity on warfarin binding. Due to experimental requirements, they only measured binding to AAG−s and showed that the binding constants depend on the different N–glycan branching resulting from glycan biosynthesis pathways. The binding constants found so far are in good agreement with our measurements, considering the different experimental techniques, although in the case of clindamycin a higher stability constant was measured in vivo and in vitro (Appendix A) [57,58,59].

## 3. Materials and Methods

### 3.1. Materials and Solution Preparation

Native human AAG (#SLCD7253) and drug compounds (clindamycin phosphate, diltiazem hydrochloride and lidocaine) were purchased from Sigma–Aldrich (Darmstadt, Germany). Warfarin sodium salt was a gift from the Department of Medicinal Chemistry, Faculty of Pharmacy and Biochemistry, Zagreb, Croatia. All chemicals were used without further purification. Immobilized SialEXO^®^ (derived from *Akkermansia muciniphilla* and expressed in *E. Coli*) was obtained from Smart Enzymes™ Genovis (Lund, Sweden), and HEPES buffer was obtained from VWR International (Leuven, Belgium). All other reagents were of analytical grade or better. All solutions were prepared immediately before the experiments in order to minimize contamination and/or impair the stability of solutions. The solutions for microcalorimetric titrations were prepared in 25 mM HEPES at pH 7.40 containing 150 mM NaCl (Kemika, Zagreb, Croatia), using doubly distilled water and 5 M NaOH (Kemika, Zagreb, Croatia) for adjusting the pH value. AAG and drug solutions were prepared by dissolving the appropriate amount of solid in the given solvent.

### 3.2. Desialylation of Human AAG 

Desialylated human AAG was prepared by incubation of immobilized sialidase beads (Immobilized SialEXO^®^), containing a resin with a mixture of two sialidases covalently coupled to agarose beads for complete removal of sialic acids (α2–3, 2–6 and 2–8, linked) of O- and N-glycosylated proteins, in the native AAG buffered stock solution (1 mg mL^−^^1^, pH 7.40, at 25 ± 1 °C). After the incubation period of one hour, the desialylated AAG is collected, washed out with buffer and concentrated by centrifugal filtration three times (1 min 1000 rcf). The complete protocol has been described by the manufacturer [60]. To confirm the characterization of desialylated AAG, samples were analyzed using the previously published method [34] to ascertain complete desialylation of AAG. Glycan composition of native and desialylated AAG variant was determined by UPLC N–glycan analysis, as described in reference [60]. The obtained N–glycans results were assigned according to the published UPLC–MS/MS procedure [34,61].

### 3.3. UV/Vis Spectrophotometry

The concentration of protein was determined spectrophotometrically (25 ± 0.1 °C) using molar extinction coefficient ($\varepsilon_{1\%}^{1\mathrm{cm}}$) of 39,200 M^−^^1^ cm^−^^1^ at 278 nm and 39,900 M^−^^1^ cm^−^^1^ at 278 nm for native and desialylated AAG, respectively, by Varian Cary 50 spectrophotometer (Melbourne, Victoria, Australia) using a quartz cell with pathlength of 1 cm (Hellma, Müllheim, Germany). HEPES was used as a reference solution. Determination of molar extinction coefficients were previously described elsewhere [62].

### 3.4. Microcalorimetric Titrations

The ITC titrations were performed on a MicroCal PEAQ–ITC calorimeter (Malvern Panalytical Ltd., Malvern, UK). The solutions were thoroughly degassed beforehand, under vacuum to avoid any formation of bubbles that might be present in the sample cell. All microcalorimetric measurements were performed at 37 °C. The volume of the sample cell was approximately 200 μL and the volume of the burette (syringe) was 42 μL. Each titration was performed in duplicate or triplicate. Blank experiments were performed to estimate corrections for the dilution heat. 

AAG solution (100–200 μM) was loaded into the sample cell using a calibrated syringe (1 mL Hamilton) and titrated with 1–2 mM drug solutions. The sample cell was continuously stirred at 700 rpm. Differential power (*DP*) is arbitrarily applied to each titration. The interval between successive injections was 200 s, which is long enough for the signal to return to its baseline and to ensure equilibrium after each addition. Double distilled water was present in the reference cell during all experiments. To correct for the thermal effect, control titrations were performed for drug solution in buffer under the same conditions as the AAG–drug titrations and dilution heat was subtracted from the AAG–drug titration data. 

All ITC data were analyzed with MicroCal PEAQ-ITC Analysis Software 1.30 using the One Set Sites Fitting Model. In order to determine whether the obtained binding parameters of the AAG–drug complex were significantly different, a two-tailed *t*-test using *p* < 0.05 threshold for statistical significance was performed. All *p* values were calculated using the *t*-test() function in Microsoft Excel (Microsoft Corporation, Redmond, WA, USA) [63].

## 4. Conclusions

AAG is a highly glycosylated protein whose plasma levels are relatively low in healthy individuals but can increase two- to six-fold in various pathophysiological conditions. The changes in plasma concentrations, as well as sialic acid content shown in this work could have important implications for drug binding to plasma proteins. Although asialo AAG is a rare condition in humans, it demonstrates the importance of the glycosylation pattern for drug binding. Further studies based on partially sialylated AAG are currently underway.

Based on calorimetric measurements by nonlinear regression using a 1:1 stoichiometric binding model (drug: protein), the binding constants of the tested drug complexes for sialylated and desialylated AAG were successfully determined. A thermodynamic study of the drug–AAG interactions showed that the enthalpic contributions to the Gibbs free energy of complex formation are favorable in all cases of the interaction (exothermic character) and that the reactions are also accompanied by a favorable increase in entropy, with the exception of warfarin. The entropic contribution to the Gibbs free energy is dominant in the complexation of three other drugs. Favorable entropic contributions are probably due to the release of solvent molecules during desolvation of the reactants, whereas favorable enthalpic contributions are due to the formation of intermolecular bonds between the functional groups of the drug and the protein. Lidocaine showed the strongest binding with AAG+s, while clindamycin showed the weakest binding with AAG−s. Importantly, removal of AAG sialic acid had the greatest effect on warfarin, so the glycosylation effect on this drug should be further pharmacokinetically investigated and eventually the dosage corrected to improve therapeutic outcome in patients with changed AAG glycosylation status.

The importance of plasma protein binding in pharmacology has been of great interest and a somewhat controversial topic in drug discovery and development over the past decades [64]. In general, highly bound drugs with low volume of distribution may have pharmacokinetic effects due to a change in protein concentration or other factors affecting binding. In this work, we have shown that sialylation affects the free drug concentration due to a change in the binding constant. Further pharmacokinetic studies are needed to demonstrate the relevance of these results and the potential application for dosage adjustments in personalized medicine. ITC is a powerful technique that provides insight into the entropy and enthalpy changes involved in the binding process in addition to binding affinity. It appears to be a relatively simple tool for screening candidate compounds for more thorough pharmacokinetic investigation. 

## Figures and Tables

**Figure 1 ijms-24-08472-f001:**
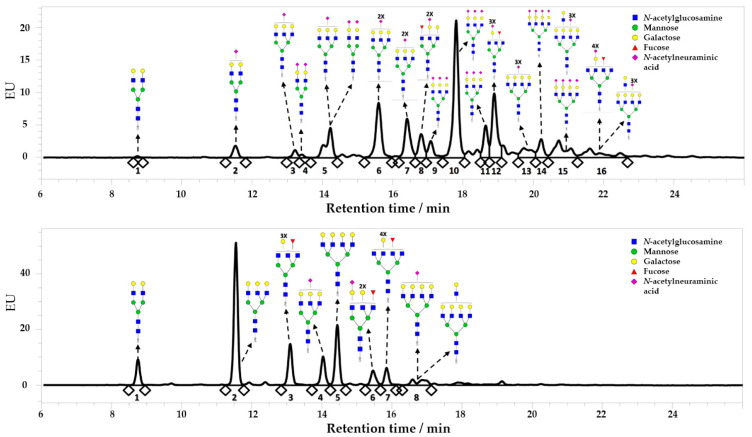
Distribution of glycan structures of AAG variants. UPLC–FLD chromatograms with structures of different fluorescently labeled N-glycans in native (top panel) and desialylated AAG (bottom panel) determined and assigned by UPLC–MS/MS using the previously published protocol [34]. The diamond shape on baseline represents the beginning and end of peak integration.

**Figure 2 ijms-24-08472-f002:**
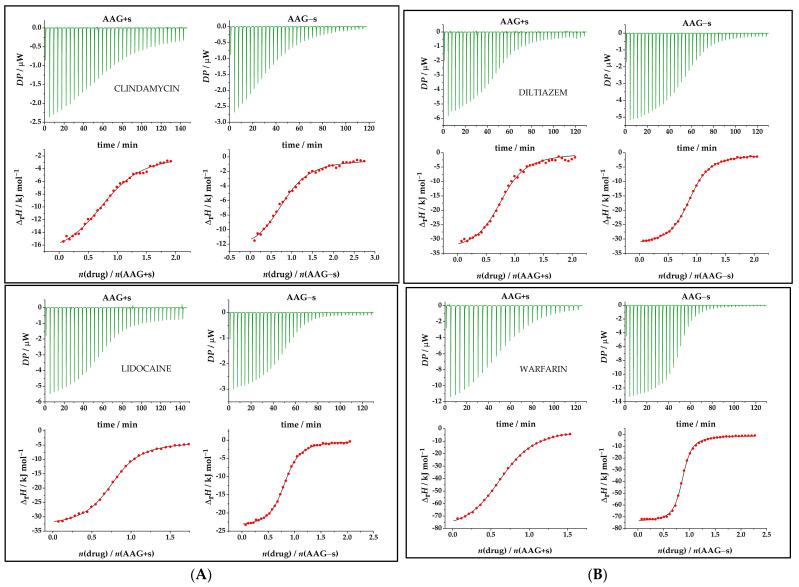
(**A**) Microcalorimetric titrations for examined drugs with native and desialylated AAG. Top charts show the differential power in μW per injectant with subtracted baseline. Bottom charts show dependence of successive enthalpy changes in kJ mol^−1^ as a function of the AAG–drug molar ratio. (**B**) Microcalorimetric titrations for examined drugs with native and desialylated AAG. Top charts show the differential power in μW per injectant with subtracted baseline. Bottom charts show dependence of successive enthalpy changes in kJ mol^−1^ as a function of the AAG–drug molar ratio. Enlarged view of injections can be seen in Appendix A. ● experimental; ―calculated.

**Figure 3 ijms-24-08472-f003:**
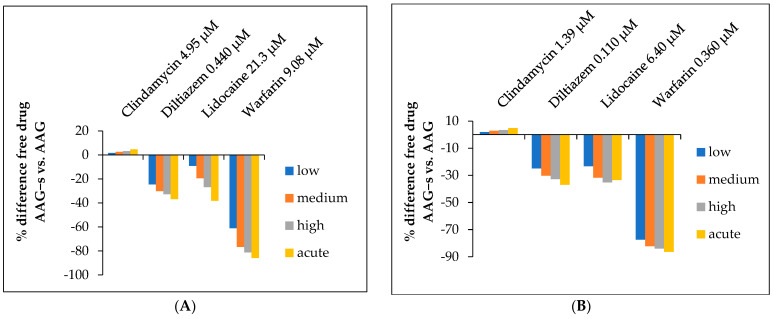
Free fraction of drugs. Percentage difference of free drug at (**A**) peak (*C*_max_), and (**B**) trough therapeutic concentrations (*C*_trough_), depending on the plasma concentrations of AAG+s or AAG−s. Values are calculated vs. the native form (AAG+s) as reference.

**Figure 4 ijms-24-08472-f004:**
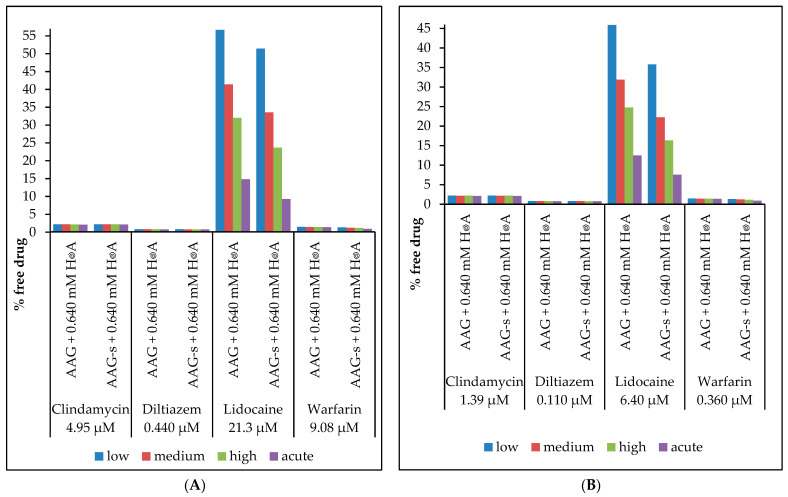
Free fraction of drugs. Percentage of free drug at (**A**) peak (*C*_max_) and (**B**) trough therapeutic concentrations (*C*_trough_), depending on the plasma concentrations of AAG+s with 0.64 mM HSA or AAG−s with 0.64 mM HSA.

**Figure 5 ijms-24-08472-f005:**
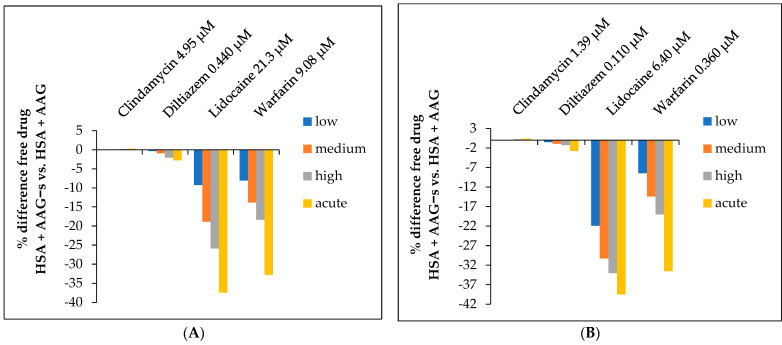
Free fraction of drugs including HSA concentration. Percentage difference of free drug at (**A**) peak (*C*_max_) and (**B**) trough therapeutic concentrations (*C*_trough_), depending on the plasma concentrations of AAG+s or AAG−s and HSA. Values are calculated vs. the native form (AAG+s) as reference.

**Table 1 ijms-24-08472-t001:** Binding affinity constants ($K_{D}$ ) for the AAG–drug interactions at pH 7.40 and 310 K.

Drug	AAG+s $K_{D}$* (µM)	AAG−s $K_{D}$* (µM)
Clindamycin	31.0 ± 2.40	32.8 ± 3.32
Diltiazem	13.2 ± 0.490	7.75 ± 1.45
Lidocaine	9.43 ± 0.950	5.21 ± 0.930
Warfarin	14.4 ± 2.83	1.65 ± 0.150

* The uncertainties are given as standard deviation of the mean (*N* ≥ 2).

**Table 2 ijms-24-08472-t002:** Thermodynamic parameters for the interaction of drugs with AAG forms at 310 K.

Drug	AAG+s			
	N *	$\Delta_{r}$*H*° * (kJ mol^−1^)	$\Delta_{r}$*S*° * (J K^−1^ mol^−1^)	$\Delta_{r}$*G*° * (kJ mol^−1^)
Clindamycin **	0.880 ± 0.169	−18.1 ± 2.48	27.9 ± 8.48	−26.8 ± 0.141
Diltiazem **	0.804 ± 0.238	−31.5 ± 4.24	7.80 ± 1.74	−29.9 ± 1.130
Lidocaine **	0.810 ± 0.0990	−29.6 ± 0.212	0.725 ± 1.53	−29.9 ± 0.283
Warfarin **	0.654 ± 0.117	−81.4 ± 1.56	−191± 5.42	−28.9 ± 0.424
	AAG−s			
Clindamycin	0.846 ± 0.0280	−23.3 ± 1.07	11.1 ± 4.42	−26.7 ± 0.283
Diltiazem	0.868 ± 0.0290	31.8 ± 1.83	4.09 ± 0.103	−30.4 ± 0.414
Lidocaine	0.818 ± 0.226	26.5 ± 3.61	15.9 ± 3.17	−31.4 ± 0.424
Warfarin	0.795 ± 0.0290	−80.1 4.31	−168 ± 5.86	−34.3 ± 0.212

* The uncertainties are given as standard deviation of the mean (*N* ≥ 2). ** p$K_{a}$ values can be seen in Appendix A.

**Table 3 ijms-24-08472-t003:** The usual therapeutic peak and trough plasma levels, *C*_max_ and *C*_trough_, for investigated drugs.

Drug	*C*_max_ (µM)	*C*_trough_ (µM)
Lidocaine ^a^	21.3	6.40
Diltiazem ^b^	0.440	0.110
Clindamycin ^c^	4.95	1.39
Warfarin ^d^	9.08	0.360

^a^ [46], ^b^ [47], ^c^ [48], ^d^ [49].

## Data Availability

All data are contained within this manuscript and the Appendix A.

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
