# Peer review of "Potential Clinically Relevant Effects of Sialylation on Human Serum AAG-Drug Interactions Assessed by Isothermal Titration Calorimetry: Insight into Pharmacoglycomics?"

_ijms, 2023, doi:10.3390/ijms24108472_

Round 1
Reviewer 1 Report
Kerep et al. studied the effects of sialylation on the binding of four representative drugs, clindamycin, diltiazem, lidocaine, and warfarin, to human serum alpha-1 acid glycoprotein. Below are my comments,
Major comments:
- In most of the negative peaks at the injection (negative deviation of the signal from baseline at each injection), the peak is consisting of more than one peaks. Could the authors provide an explanation for it?
- Could the authors provide docking results of the drugs to better understand the effects of sialylation on drug binding?
Minor comments:
- “Relatively small differences in the ∆rG° and the ∆rH° values indicate a dominant effect of enthalpy on binding” -> “Relatively small differences in the ∆rG° and the ∆rH° values indicate a dominant effect of entropy on binding”
Author Response
My colleagues and I are grateful for the editor's and reviewers' interest in our manuscript and for the constructive criticism they have offered. Below is a point-by-point response to the reviewers.
Please note that all changes in the revised manuscript have been highlighted in yellow for clarity.
- In most of the negative peaks at the injection (negative deviation of the signal from baseline at each injection), the peak is consisting of more than one peaks. Could the authors provide an explanation for it?
Answer: There are no additional peaks for any of the injections, but it may appear so due to the line thickness in the presented thermograms. For clarity, we have provided higher resolution images in the manuscript (Figs 2A,B) and additional thermogram in supporting information (Fig. S6) for better overview.
- Could the authors provide docking results of the drugs to better understand the effects of sialylation on drug binding?
Answer: We thank the reviewer for this suggestion, and we agree that the molecular docking study would be a valuable contribution to this paper. However, we think the reported results are significant per se and such additional study may be attempted in the future. Furthermore, the appropriate crystal structure of desialylated AAG is not available in the PDB and therefore the molecular docking cannot be performed with desialylated AAG.
- “Relatively small differences in the ∆rG° and the ∆rH° values indicate a dominant effect of enthalpy on binding” -> “Relatively small differences in the ∆rG° and the ∆rH° values indicate a dominant effect of entropy on binding”
Answer: corrected in manuscript
Reviewer 2 Report
This paper investigates binding of several drugs to protein AAG by using isothermal titration calorimetry. Titration data appears to be solid. If one wants to be nit-picking, weak binding curve for clindamycin (AAG+s) is not reached to saturation. This is because optimal C values are between 1 and 10. However, the data appears to be reasonable. I agree with some of the interpretation of the data, but some aspects need more development. For example:
Line 177: The authors appear to interpret entropy changes only in terms of structure. Even if this were true, the suggested possibility for increased entropy of lidocaine binding seems to be counter intuitive. The authors attribute this to the flexibility of the drug. One would think that upon binding it would become more rigid, hence loss of entropy. Similar suggestions, attributing the entropy to only structural aspects of the protein and drug, are given in other places too (lines 190 and starting from 193). One should keep in mind that binding of ligands free up tightly-bound water molecules that would gain entropy. Therefore, the authors should revise the manuscript to discuss other possibilities like this for entropic contributions. Moreover, similarity of pKa values were also mentioned. pKa values can change upon binding and that may not be the same for all cases. One way to address to see any net change in charge is to perform titrations in buffers with different heats of ionization to determine so called ‘delta n’ value, which is indicator of the net protonation or deprotonation.
If the commercially used AAG is a mixture of variants shown in figure 1, the observed results are really a weighted average of binding to these variants. How this may influence some ofthe conclusions?
It is also unclear to me what is the reason for using 0.64 mM albumin (while AAC is at much lower concentration)? Albumin is known to be a pretty “sticky” and binds a lot of molecules. Its presence in large excess may have some effects.
Is the
I am not sure I understand the rationale for the selection of these drugs for study. Authors should give reason(s). I was expecting a bit more specific conclusions.
Overall, there is a need for revision before consideration for publication.
Author Response
My colleagues and I are grateful for the editor's and reviewers' interest in our manuscript and for the constructive criticism they have offered. Below is a point-by-point response to the reviewers.
Please note that all changes in the revised manuscript have been highlighted in yellow for clarity.
1. Line 177: The authors appear to interpret entropy changes only in terms of structure. Even if this were true, the suggested possibility for increased entropy of lidocaine binding seems to be counter intuitive. The authors attribute this to the flexibility of the drug. One would think that upon binding it would become more rigid, hence loss of entropy. Similar suggestions, attributing the entropy to only structural aspects of the protein and drug, are given in other places too (lines 190 and starting from 193). One should keep in mind that binding of ligands free up tightly-bound water molecules that would gain entropy. Therefore, the authors should revise the manuscript to discuss other possibilities like this for entropic contributions. Moreover, similarity of pKa values were also mentioned. pKa values can change upon binding and that may not be the same for all cases. One way to address to see any net change in charge is to perform titrations in buffers with different heats of ionization to determine so called ‘delta n’ value, which is indicator of the net protonation or deprotonation.
Answer: We thank the reviewer for these comments. We have added more discussion for entropic contributions in binding in the Results and Discussion section starting at lines 176 and 188 (highlighted) and from 205 to 210. We have also added the possibility of determining the number of protons exchanged for different buffers starting at line 211.
- If the commercially used AAG is a mixture of variants shown in figure 1, the observed results are really a weighted average of binding to these variants. How this may influence some of the conclusions?
Answer: Commercially available AAG is highly glycosylated, with sialic acids accounting for as much as 11% of the total mass. While it is entirely possible that the mentioned variants have an influence, we always used the same standard for all experiments before and after removal of the sialic acids. Therefore, any influence observed should come only from the difference in sialic acid content, since the measurements are always performed with the same combination of variants, i.e., the same commercial sample with and without sialic acids (approximately 94% of sialic acids were removed in AAG-s). It is undeniable that it would be ideal to have an AAG sample with specific isolated glycoforms, but this would be very difficult to perform, especially considering that AAG has up to five glycosylation sites which results in high microheterogeneity. Such an additional study could be attempted in the future, but we think it is beyond the scope of this paper. We have commented on this in the Results and Discussion section starting at line 126 (highlighted).
- It is also unclear to me what is the reason for using 0.64 mM albumin (while AAC is at much lower concentration)? Albumin is known to be a pretty “sticky” and binds a lot of molecules. Its presence in large excess may have some effects.
Answer: It is true that HSA binds a variety of molecules, and it is for this very reason that we also considered the mean physiological concentration of albumin in healthy humans to obtain more realistic estimates of free drug concentrations when considering drug binding to AAG. Since the level for albumin (66 500 Da) in human serum ranges from 30-50 mg/mL, we obtain 42.5 mg/mL as an average value, which corresponds to 0.64 mM used for the calculation of the free drug concentration. We thank the reviewer for this comment and have rewritten a short section in the discussion on the use of HSA starting at line 244.
- I am not sure I understand the rationale for the selection of these drugs for study. Authors should give reason(s). I was expecting a bit more specific conclusions.
Answer: We have added an explanation for the use of selected drugs in the Results and Discussion section starting at line 88 and we have also added a segment in the Conclusion section between line 364 and 379.
Round 2
Reviewer 2 Report
The authors responded to my comments with reasonable edits and rewrites, so I have no further comments